# PAC-Bayesian Neural Network Bounds

## Abstract

Bayesian neural networks, which both use the negative log-likelihood loss function and average their predictions using a learned posterior over the parameters, have been used successfully across many scientific fields, partly due to their ability to 'effortlessly' extract desired representations from many large-scale datasets. However, generalization bounds for this setting is still missing. In this paper, we present a new PAC-Bayesian generalization bound for the negative log-likelihood loss which utilizes the *Herbst Argument* for the log-Sobolev inequality to bound the moment generating function of the learners risk. We explore the generalization and calibration properties of the learned posterior on several image classification benchmarks, showing that the proposed approach provides better generalization and uncertainty estimates.

## 1 Introduction

Deep neural networks are ubiquitous across disciplines and often achieve state of the art results (e.g., Krizhevsky et al. (2012); Simonyan & Zisserman (2014); He et al. (2016)). Albeit neural networks are able to encode highly complex input-output relations, in practice, they do not tend to overfit (Zhang et al., 2016). This tendency to not overfit has been investigated in numerous works on generalization bounds (Langford & Shawe-Taylor, 2002; Langford & Caruana, 2002; Bartlett et al., 2017a; 2019; McAllester, 2003; Germain et al., 2016; Dziugaite & Roy, 2017). Indeed, many generalization bounds apply to neural networks. However, most of these bounds assume that the loss function is bounded (Bartlett et al., 2017a; Neyshabur et al., 2017; Dziugaite & Roy, 2017). Unfortunately, this assumption excludes the popular negative log-likelihood (NLL) loss, which is instrumental to Bayesian neural networks that have been used extensively to calibrate model performance and provide uncertainty measures to the model prediction.

In this work we introduce a new PAC-Bayesian generalization bound for NLL loss of deep neural networks. Our work utilizes the *Herbst argument* for the *logarithmic-Sobolev* inequality (Ledoux, 1999) in order to bound the moment-generating function of the model risk. Broadly, our PAC-Bayesian bound is comprised of two terms: The first term is dominated by the norm of the gradients with respect to the input and it describes the expressivity of the model over the prior distribution. The second term is the KL-divergence between the learned posterior and the prior, and it measures the complexity of the learning process. In contrast, bounds for linear models or bounded loss functions lack the term that corresponds to the expressivity of the model over the prior distribution and therefore are the same when applied to shallow and deep models.

We empirically show that our PAC-Bayesian bound is tightest when we learn the mean and variance of each parameter separately, as suggested by Blundell et al. (2015) in the context of Bayesian neural networks (BNNs). We also show that the proposed bound holds different insights regarding model architecture, optimization and prior distribution selection. We demonstrate that such optimization minimizes the gap between risk and the empirical risk compared to the standard Bernoulli dropout and other Bayesian inference approximation while being consistent with the theoretical findings. Additionally, we explore in-distribution and out-of-distribution examples to show that such optimization produces better uncertainty estimates than the baseline.

## 2 RELATED WORK

Generalization bounds for neural networks were explored in various settings. VC-theory provides both upper bounds and lower bounds to the network VC-dimension, which are linear in the number of network parameters (Bartlett et al., 2017b; 2019). While VC-theory asserts that such a model should overfit as it can learn any random labeling (e.g., Zhang et al. (2016)), surprisingly, neural networks generally do not overfit. Rademacher complexity allows to apply data dependent bounds to neural networks (Bartlett & Mendelson, 2002). However, they rely on bounds of the loss function (Wan et al., 2013; Gao & Zhou, 2016). Rademacher complexity bounds may also take into account the Lipschitz constant of the network. While this presents an improvement when its gradient-norms are bounded, these results also require the loss to be bounded. Stability bounds may be applied to unbounded loss functions and in particular to the negative log-likelihood (NLL) loss (Bousquet & Elisseeff, 2002; Rakhlin et al., 2005; Shalev-Shwartz et al., 2009; Hardt et al., 2015; Zhang et al., 2016). However, stability bounds for convex loss functions, e.g., for logistic regression, do not apply to neural networks since they require the NLL loss to be a convex function of the parameters. Alternatively, Hardt et al. (2015) estimate the stability of stochastic gradient descent dynamics, which strongly relies on early stopping. This approach results in weaker bounds for the non-convex setting.

PAC-Bayesian bounds were recently applied to neural networks (Langford & Caruana, 2002; McAllester, 2013; Dziugaite & Roy, 2017; Neyshabur et al., 2017). In contrast to our work, those related works all consider bounded loss functions. PAC-Bayesian bounds for the NLL loss in the online setting were put forward by Banerjee (2006). The online setting does not to consider the whole sample space and therefore is simpler to analyze in the Bayesian setting. An excellent survey on PAC-Bayesian bounds was provided by Germain et al. (2016). Alquier et al. (2016) introduced PAC-Bayesian bounds for linear classifiers with the hinge-loss. Germain et al. (2016) is closer to our setting and considers PAC-Bayesian bounds for the quadratic loss function, while proving that the quadratic loss function has exponential decay (sub-gamma) when the data is sampled from a Gaussian distribution. This result is a PAC-Bayesian generalization bound for the NLL loss of a regressor which assumes Gaussian model over a continuous label space $y$. Our work differs from this work in two important aspects: (1) our work does not consider linear classifiers and apply to any deep neural networks; (2) our bound considers the NLL loss for classification tasks, for which the label space $y$ is discrete.

PAC-Bayesian bounds for the NLL loss function are intimately related to learning Bayesian inference (Germain et al., 2016). Recently many works applied various posteriors in Bayesian neural networks. Gal & Ghahramani (2015); Gal (2016) introduce a Bayesian inference approximation using Monte Carlo (MC) dropout, which approximates a Gaussian posterior using Bernoulli dropout. Srivastava et al. (2014) introduced Gaussian dropout which effectively creates a Gaussian posterior that couples between the mean and the variance of the learned parameters. Kingma et al. (2015) explored the relation of this posterior to log-uniform priors, while Blundell et al. (2015) suggests to take a full Bayesian perspective and learn separately the mean and the variance of each parameter. Our work uses the bridge between PAC-Bayesian bounds and Bayesian inference, as described by Germain et al. (2016), to find the optimal prior parameters in PAC-Bayesian setting and apply it in the Bayesian setting.

Most of the literature regarding Bayesian modeling involves around a two-step formalism (Bernardo & Smith, 2009): (1) a prior is specified for the parameters of the deep net; (2) given the training data, the posterior distribution over the parameters is computed and used to quantify predictive uncertainty. Since exact Bayesian inference is computationally intractable for neural networks, approximations are used, including MacKay (1992); Hernández-Lobato & Adams (2015); Hasenclever et al. (2017); Balan et al. (2015); Springenberg et al. (2016). In this study we follow this two-step formalism, particularly we follow a similar approach to Blundell et al. (2015) in which we learn the mean and standard deviation for each parameter of the model using variational Bayesian practice. Our experimental validation emphasizes the importance of learning both the mean and the variance.

## 3 BACKGROUND

Generalization bounds provide statistical guarantees on learning algorithms. They measure how the learned parameters $w$ perform on test data given their performance on the training data $S =$

$\{(x_1, y_1), \ldots, (x_m, y_m)\}$, where $x_i$ is the data instance and $y_i$ is its corresponding label. The performance of the learning algorithm is measured by a loss function $\ell(w, x, y)$. The risk of a learner is its average loss, when the data instance and its label are sampled from their true but unknown distribution $D$. We denote the risk by $L_D(w) = \mathbb{E}_{(x,y) \sim D} \ell(w, x, y)$. The empirical risk is the average training set loss $L_S(w) = \frac{1}{m} \sum_{i=1}^{m} \ell(w, x_i, y_i)$.

PAC-Bayesian theory bounds the risk of a learner $\mathbb{E}_{w \sim q} L_D(w)$ when the parameters are averaged over the learned posterior distribution $q$. The parameters of the posterior distribution are learned from the training data $S$. In our work we focus on the following PAC-Bayesian bound:

**Theorem 1** (Alquier et al. (2016)). *Let $KL(q||p) = \int q(w) \log(q(w)/p(w)) dw$ be the KL-divergence between two probability density functions $p, q$. For any $\lambda > 0$ and for any $\delta \in (0, 1]$ and for any prior distribution $p$, with probability at least $1 - \delta$ over the draw of the training set $S$, the following holds simultaneously for any posterior distribution $q$:*

$$\mathbb{E}_{w \sim q}[L_D(w)] \leq \mathbb{E}_{w \sim q}[L_S(w)] + \frac{\log \mathbb{E}_{w \sim p, S \sim D^m}[e^{\lambda(L_D(w) - L_S(w))}] + KL(q||p) + \log(1/\delta)}{\lambda}. \quad (1)$$

PAC-Bayesian theory is intimately connected to Bayesian inference when considering the negative log-likelihood loss function $\ell(w, x, y) = -\log p(y|x, w)$ and $\lambda = m$. Germain et al. (2016) proved that the optimal posterior in this setting is $q(w) = p(w|S)$. Bayesian inference considers the posterior $p(y|x, S) = \int p(w|S) p(y|x, w) dw$, at test time for a data instance $x$, which corresponds to the risk of the optimal posterior. Unfortunately, the optimal posterior is rarely available, and PAC-Bayes relies on the approximated posterior $q$.

Coincidently, the approximated posterior and its KL-divergence from the prior distribution are instrumental to the evidence lower bound (ELBO), which is extensively used in Bayesian neural networks (BNNs) to bound the log-likelihood $\sum_{i=1}^{m} \log p(y_i|x_i)$:

$$-\sum_{i=1}^{m} \log p(y_i|x_i) \leq -\sum_{i=1}^{m} \mathbb{E}_{w \sim q} \log p(y_i|x_i, w) + KL(q||p) \quad (2)$$

While the right hand side of a PAC-Bayesian bound, with the negative log-likelihood loss and $\lambda = m$, is identical to the right hand side of the ELBO bound in term of learning, they serve different purposes. One is used for bounding the risk while the other is used for bounding the marginal log-likelihood. Nevertheless, the same algorithms can be used to optimize BNNs and PAC-Bayesian intuitions and components can influence the practice of Bayesian neural networks.

## 4 PAC-BAYESIAN BOUNDS FOR THE NEGATIVE LOG-LIKELIHOOD LOSS

It is challenging to derive a PAC-Bayesian bound for the negative log-likelihood (NLL) loss as it requires a bound on the log-partition function $\log \mathbb{E}_{w \sim p, S \sim D^m}[e^{\lambda(L_D(w) - L_S(w))}]$. In cases where the loss function is uniformly bounded by a constant, e.g., the zero-one loss, the log-partition function is bounded as well. Unfortunately, the NLL loss is unbounded, even when $y$ is discrete. For instance, consider fully connected case, where the input vector of the $(k)$-th layer is a function of the parameters of all previous layers, i.e., $x_k(W_0, \ldots, W_{k-1})$. The entries of $x_k$ are computed from the response of its preceding layer, i.e., $W_{k-1} x_k$, followed by a transfer function $\sigma(\cdot)$, i.e., $x_{k+1} = \sigma(W_{k-1} x_k)$. Then, since the NLL is define as $-\log(p(y|x, w)) = -(W_k x_k)_y + \log(\sum_{\hat{y}} e^{(W_k x_k)_{\hat{y}}})$, if the rows in $W_k$ consist of the vector $r x_k$ then the NLL loss increases with $r$, and is unbounded when $r \to \infty$.

Our main theorem shows that for smooth loss functions, the log-partition function is bounded by the expansion of the loss function, i.e., the norm of its gradient with respect to the data $x$. This property is appealing since these gradients often decay rapidly for deep neural networks, as we demonstrate in our experimental evaluation. Consequently deep networks enjoy tighter generalization bounds than shallow networks. Our proof technique follows the Herbst Argument for bounding the log-partition function using the Log-Sobolev inequality for Gaussian distributions (Ledoux, 2001).

**Theorem 2.** *Assume $(x, y) \sim D$ and $x$ given $y$ follows the Gaussian distribution. Let $\ell(w, x, y)$ be a smooth loss function (e.g., the negative log-likelihood loss). For any $\delta \in (0, 1]$ and for any real number $\lambda > 0$, with probability at least $1 - \delta$ over the draw of the training set $S$ the following holds*

*simultaneously for any posterior probability density function:* $\mathbb{E}_{w \sim q}[L_D(w)] \leq$

$$\mathbb{E}_{w \sim q}[L_S(w)] + \frac{\log \mathbb{E}_{w \sim p} e^{\frac{2\lambda^2}{m} \mathbb{E}_{(x,y) \sim D}\left[\|\nabla_x \ell(w,x,y)\|^2 \int_0^1 \frac{e^{-\alpha \ell(w,x,y)}}{\mathbb{E}_{(\hat{x},\hat{y}) \sim D} e^{\alpha(-\ell(w,\hat{x},\hat{y}))}} d\alpha\right]}}{\lambda} + KL(q||p) + \log(1/\delta)}{\lambda}.$$

(3)

The Gaussian assumption for the data generating distribution $D$ can be relaxed to any log-concave distribution, using Gentil (2005), Corollary 2.5. We use the Gaussian assumption to avoid notational overhead.

Broadly, the proposed bound is comprised of two terms: The first term is the log-partition function which is dominated by the norm of the gradients with respect to the input, namely $\frac{2\lambda^2}{m} \mathbb{E}_{(x,y) \sim D}\left[\|\nabla_x \ell(w,x,y)\|^2 \int_0^1 \frac{e^{-\alpha \ell(w,x,y)}}{\mathbb{E}_{(\hat{x},\hat{y}) \sim D} e^{\alpha(-\ell(w,\hat{x},\hat{y}))}} d\alpha\right]$, and it describes the expressivity of the model over the prior distribution. The second term is the KL-divergence between the learned posterior and the prior, and it measures the complexity of the learning process.

The proof starts with Eq. (1) and uses the Herbst Argument and the Log-Sobolev inequality to bound the moment-generating function $\mathbb{E}_{S \sim D^m}[e^{\lambda(L_D(w) - L_S(w))}]$. Specifically, the proof consists of three steps. First we use the statistical independence of the training samples to decompose the moment generating function

$$\mathbb{E}_{S \sim D^m}[e^{\lambda(L_D(w) - L_S(w))}] = e^{\lambda \mathbb{E}_{(x,y) \sim D} \ell(w,x,y)} \left(\mathbb{E}_{(\hat{x},\hat{y}) \sim D}[e^{\frac{\lambda}{m}(-\ell(w,\hat{x},\hat{y}))}]\right)^m.$$

(4)

Then we use the Herbst argument to bound the function $M\left(\frac{\lambda}{m}\right) \triangleq \mathbb{E}_{(\hat{x},\hat{y}) \sim D}[e^{\frac{\lambda}{m}(-\ell(w,\hat{x},\hat{y}))}]$ and obtain the following bound:

$$M\left(\frac{\lambda}{m}\right) = e^{\log M(0) + \left(\frac{\lambda}{m}\right)^2 \int_0^1 \frac{\alpha M'(\alpha) - M(\alpha) \log M(\alpha)}{\alpha^2 M(\alpha)} d\alpha}.$$

(5)

Finally we use the log-Sobolev inequality for Guasslev distributions,

$$\alpha M'(\alpha) - M(\alpha) \log M(\alpha) \leq 2 \cdot \mathbb{E}_{(x,y) \sim D}[e^{-\alpha \ell(w,x,y)} \alpha^2 \|\nabla_x \ell(w,x,y)\|^2].$$

(6)

The above theorem can be extended to settings for which $x$ is sampled from any log-concave distribution, e.g., the Laplace distribution. The log-concave setting modifies the gradient norm and the log-Sobolev constant 2 in Eq. (6) that corresponds to Gaussian distributions, cf. Gentil (2005). We avoid this generalization to simplify our mathematical derivations.

A detailed description of the proof can be found on Section 8.1 in the Appendix.

## 5 APPLICATION TO LOGISTIC REGRESSION

The bound in Theorem 2 is favorable when applied to deep networks since their gradients w.r.t. data often decay rapidly. Nevertheless we can also apply our technique to shallow nets trained with NLL loss. We obtain PAC-Bayesian bounds for multi-class logistic regression.

The NLL loss for multiclass logistic regression takes the form: $-\log p(y|x,w) = -(Wx)_y + \log(\sum_{\hat{y}} e^{(Wx)_{\hat{y}}})$, where $x \in \mathbb{R}^d$ is the data instance, $y \in \{1, \ldots, k\}$ are the possible labels, and $W \in \mathbb{R}^{k \times d}$ is the matrix of parameters. The bound in Theorem 2 takes the form:

**Corollary 1.** *Assume* $(x,y) \sim D$ *and* $x \in \mathbb{R}^d$ *given* $y$ *follows the Gaussian distribution. Let* $\ell(w,x,y) = -\log p(y|x,w)$ *be the negative log-likelihood loss for* $k-$*class logistic regression. For any* $\delta \in (0,1]$*, for any* $\lambda > 0$ *and for any prior density function with variance* $\sigma_p^2 \leq m/16\lambda^2$*, with probability at least* $1 - \delta$ *over the draw of the training set* $S$ *the following holds simultaneously for any posterior probability density function:*

$$\mathbb{E}_{w \sim q}[L_D(w)] \leq \mathbb{E}_{w \sim q}[L_S(w)] + \frac{kd \log(2) + KL(q||p) + \log(1/\delta)}{\lambda}.$$

(7)

Full proof can be found on Section 8.2 in the Appendix, while we sketch the main steps of the proof below. The above corollary shows that PAC-Bayesian bound for classification using the NLL loss

Table 1: Different $\sigma_p$ values experiments. All models composed of five fully connected layers with ReLU activation functions.

|  | Model | Train loss | Test loss | Train acc. | Test acc. | Gen. loss | Bound |
|---|---|---|---|---|---|---|---|
| MNIST | $\sigma_p = 0.05$ | 0.07 | 0.08 | 98.48 | 97.73 | 0.02 | 0.2 |
|  | $\sigma_p = 0.1$ | **0.02** | **0.06** | **99.53** | **98.27** | 0.03 | **0.08** |
|  | $\sigma_p = 0.2$ | 0.03 | 0.08 | 99.24 | 97.9 | 0.05 | **0.08** |
|  | $\sigma_p = 0.3$ | 0.09 | 0.11 | 97.2 | 96.66 | 0.02 | 0.67 |
| Fashion | $\sigma_p = 0.05$ | 0.24 | 0.32 | 92.03 | 88.55 | 0.09 | 0.2 |
|  | $\sigma_p = 0.1$ | **0.11** | **0.25** | **93.28** | **89.50** | 0.1 | **0.13** |
|  | $\sigma_p = 0.2$ | 0.26 | 0.33 | 90.12 | 87.87 | 0.07 | 0.86 |
|  | $\sigma_p = 0.3$ | 0.37 | 0.42 | 86.36 | 84.26 | 0.05 | 21.73 |

can achieve rate of $\lambda = m$. This result augments the PAC-Bayesian for regression using the NLL loss for regression, i.e., the square loss, of Germain et al. (2016).

The PAC-Bayesian bound for logistic regression is derived by applying Theorem 2. We begin by realizing the gradient of $\log p(y|x, w)$ with respect to $x$. We denote by $w_y$ the $y-$th row of the parameter matrix $W$. Thus $\nabla_x \log p(y|w, x) = \sum_{\hat{y}} p(\hat{y}|x, w)(w_y - w_{\hat{y}})$, and the gradient norm is upper bounded as follows: $\|\nabla_x \log p(y|w, x)\|^2 \leq 2 \sum_y \|w_y\|^2$. Plugging this result into Eq. (18) we obtain the following bound:

$$\mathbb{E}_{w \sim p, S \sim D^m}\left[e^{\lambda(L_D(w) - L_S(w))}\right] \quad \leq \quad \mathbb{E}_{w \sim p} e^{\frac{4\lambda^2}{m} \sum_{\hat{y}} \|w_{\hat{y}}\|^2}. \tag{8}$$

Finally, whenever $\lambda \sigma_p \leq \sqrt{m/8}$ we derive the bound

$$\mathbb{E}_{w \sim p} e^{\frac{4\lambda^2}{m} \sum_{\hat{y}} \|w_{\hat{y}}\|^2} \leq \left(\frac{m}{m - 8\lambda^2 \sigma_p^2}\right)^{kd} \tag{9}$$

A detailed description of the proof can be found on Section 8.2 in the Appendix.

## 6 EXPERIMENTS

In this section we study the derived bound empirically. We start with an ablation study of the proposed bound using classification and regression models. Next, we present our results for multi-class classification tasks using different datasets and different architectures. We conclude the section, with an analysis of the models' uncertainty estimates using for in-distribution examples and out-of-distribution examples. All suggested models follows the a Bayesian Neural Networks (BNN) perspective, in which we learn the mean and standard deviation for each learnable parameter in the network where we define $N(0, \sigma_p^2 I)$ to be the prior over weights.

### 6.1 ABLATION

**Effect of $\sigma_p$.** We start by exploring the effect of $\sigma_p$ on the models' performance and the proposed generalization bound. For that, we trained several models using $\sigma_p \in \{0.05, 0.1, 0.2, 0.3\}$ using the MNIST (LeCun & Cortes, 2010) and Fashioin-MNIST Xiao et al. (2017) datasets. All results were obtained using fully connected layers with ReLU as non-linear activation function. We optimized the NLL loss function using Stochastic Gradient Descent (SGD) for 50 epochs with a learning rate of 0.01 and momentum of 0.9. For each model we compute the average train and test loss and accuracy together with the absolute difference between the training loss and the test loss, denoted as *Generalization Loss*. Moreover, we compute the generalization bound as stated in Eq. (18) for all settings. Results are summarized in Table 1.

Although $\sigma_p = 0.2$ reaches slightly better generalization bound on MNIST dataset, $\sigma_p = 0.1$ performs better over all calculated metrics, i.e., average loss and accuracy, both on MNIST and Fashion-MNIST. Notice, for Fashion-MNIST we observed slightly better generalization gap while using $\sigma_p = 0.05$, however, its loss and accuracy are worse comparing to $\sigma_p = 0.1$.

**Effect of $\lambda$.** Recall, we bound the moment generating function using the norm of the functions' gradient with respect to the data $x$ (Eq. (18)). To construct tighter generalization bounds, we would like to set $\lambda \to m$. However, in Eq. (18) $\lambda$ appears in both numerator and denominator. It is

Table 2: Different depth level experiments. For fair comparison all models have roughly the same number of parameters (~80K for MNIST, ~800K for Fashion-MNIST, ~1500 for regression). All models composed of fully connected layers with ReLU activation function.

| | Model | Train loss | Test loss | Train acc. | Test acc. | Gen. loss | Bound |
|---|---|---|---|---|---|---|---|
| MNIST | One layer | 0.253 | 0.267 | 92.89 | 92.49 | 0.015 | 3.69 |
| | Two layers | 0.045 | 0.073 | 98.89 | 97.82 | 0.028 | 2.28 |
| | Three layers | 0.029 | 0.063 | 99.41 | 98.15 | **0.033** | 0.82 |
| | Four layers | 0.024 | 0.065 | 99.49 | 98.14 | 0.041 | 0.13 |
| | Five layers | **0.023** | **0.058** | **99.52** | **98.27** | 0.035 | **0.08** |
| Fashion | One layer | 0.38 | 0.46 | 86.86 | 84.2 | 0.07 | 5.68 |
| | Two layers | 0.21 | 0.31 | 92.52 | 88.9 | 0.1 | 4.9 |
| | Three layers | 0.21 | 0.31 | 92.34 | 88.7 | 0.1 | 1.08 |
| | Four layers | 0.21 | 0.30 | 92.49 | 88.8 | **0.09** | 0.22 |
| | Five layers | **0.19** | **0.29** | **93.3** | **89.6** | 0.1 | **0.13** |
| Boston | One layer | 45.7 | 106.5 | - | - | 60.8 | 378.7 |
| | Two layer | 8.6 | 15.7 | - | - | 7.1 | 286.3 |
| | Three layers | 8.2 | 12.9 | - | - | 4.7 | 105.3 |
| | Four layers | 10.4 | 14.1 | - | - | 3.7 | 45.9 |
| | Five layers | 12.0 | **12.1** | - | - | **0.1** | **31.9** |

hence not clear whether the bound will converge, which depends on the model architecture, which is represented by the norm of its gradient. In other words, models with lower gradient norm could benefit from larger values of $\lambda$, hence tighter generalization bounds.

To further explore this property we trained five different models with different number of layers (1-5). We look into both classification models while optimizing the NLL loss function, and regression tasks while optimizing the Mean Squared Error (MSE) loss function. For classification we used MNIST and Fashion-MNIST datasets, while for regression we use the Boston Housing dataset (for the regression models, results were obtained using 5-fold cross validation). Except for the linear models, we force all models to have roughly the same numbers of parameters (~80K for MNIST, ~800K for Fashion-MNIST, ~1500 for regression). For all models we set ReLU as non-linear activation functions. We optimize all models for 50 epochs using SGD with learning rate of 0.01 and momentum of 0.9. Based on results of the prior paragraph, in all reported settings we set $\sigma_p = 0.1$.

Results are reported in Table 2. It can be seen that deeper models produce tighter generalization bounds on all three datasets. When considering model performance on down-stream classification task we notice that in general, models with better generalization bounds perform slightly better in terms of loss and accuracy. One possible explanation is that deeper models have smaller gradients w.r.t. the input. To validate that we further computed the average squared gradient norm w.r.t. the input as a function of the model depth, for both MNIST and Fashion-MNIST datasets. It can be seen from Figure 1a that indeed the gradients decay rapidly as we add more layers to the network.

Next, we present in Figure 1b the generalization bound as a function of $\lambda$ for MNIST models. We explored $\lambda \in [\sqrt{m}, m]$ and stopped the plot once the bound can no longer be computed. Experiments using Fashion-MNIST produce similar plot and can be found on Section 8.7 in the Appendix.

**Weights visualization.** Since we consider Bayesian Neural Networks (BNNs) and optimize the KL-divergence between the prior and the posterior over the weights, we can visualize the average mean and standard deviation (STD) of the posterior as a function of the model depth. Figure 2a presents this for MNIST and Fashion-MNIST models using four and five depth levels. As expected, we can see that the average mean over the weights is zero for all layers while weights STD approaches 0.1. For the MNIST models (Figure 2a top row), we observed the standard deviation are ~0.7 and not 0.1. We suspect this behaviour is due to fast optimization, hence the models do not have much signal to push the model towards the prior distribution. Notice, in all settings the average STD of the model weights decreases on the last layer. We observed a similar behavior also for the other models.

## 6.2 CLASSIFICATION

Next, we compare BNN models against two commonly used baselines. The first baseline is a soft-max model using the same architecture as the BNN while adding dropout layers. The second base-

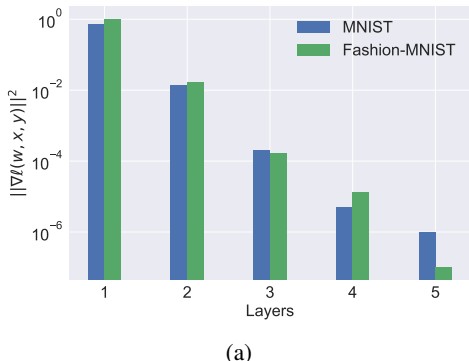 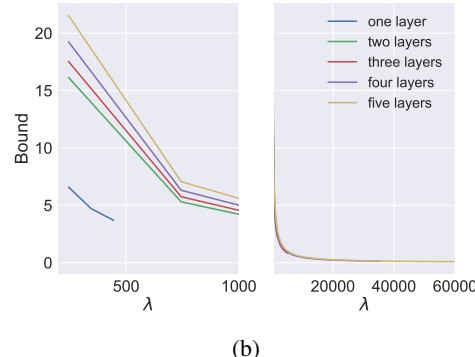

(a)                 (b)

Figure 1: Analysis of the proposed bound as a function network depth. In (a) we show the squared gradient norm as a function of the layers of the model. In (b) we report the the generalization bound as a function of $\lambda$ for different deep net depth levels using the MNIST dataset.

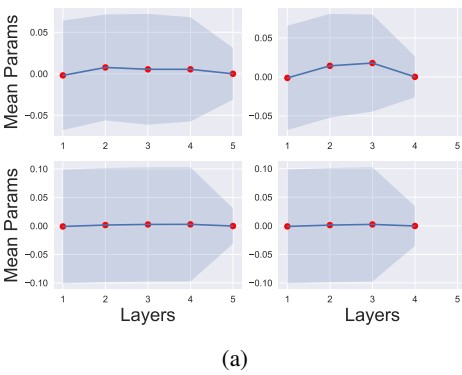 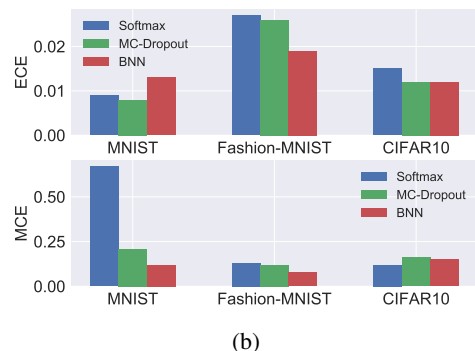

(a)                 (b)

Figure 2: In (a): average mean and STD of model weights as a function of depth level. Top row is for MNIST, bottom row is for Fashion-MNIST. In (b): ECE and MCE metrics for Softmax, MC-Dropout and BNN models using MNIST, Fashion-MNIST, and CIFAR10 (lower is better).

line is a Bayesian approximation using Monte Carlo Dropout (Gal & Ghahramani, 2015), denoted as MC-Dropout, using different dropout rates and weight decay value of 1e-5.

To evaluate these approaches we conducted multi-class classification experiments using three classification benchmarks: MNIST, Fashion-MNIST, and CIFAR-10 (Krizhevsky & Hinton, 2009). We report train set and test set loss and accuracy, together with their generalization gaps (e.g., the difference between the test and training loss and accuracy). Notice, as oppose to Dziugaite & Roy (2017) our results are reported for multi-class classification and not for binary classification. For completeness, we report binary classification results on Section 8.4 in the Appendix. The premise beyond these type of experiments is to preset the benefit of learning the mean and STD separately for each of the models' parameters.

Results are reported in Table 7. For BNN and MC-Dropout models, we sample 20 times from the posterior distribution and average their outputs to produce the model output. We also sampled more times, however, we did not see any significant differences. We observe that BNN models achieve comparable results to both baselines but with lower loss and accuracy generalization gaps. Throughout the experiments we use dropout value of 0.3 for Softmax and MC-Dropout models and $\sigma_p = 0.1$ for BNN models. We chose these values after grid search over different dropout values for all baseline models. A detailed description of all implementation details together with results for more dropout rates can be found on Sections 8.4, 8.5, and 8.3 in the Appendix.

### 6.3 UNCERTAINTY ESTIMATES

Lastly, we evaluated the uncertainty estimates of BNN models against softmax models and MC-Dropout models. We experimented with both in-distribution and out-of-distribution examples. The purpose of the following experiments is to demonstrate that following the Bayesian approach together with the carefully picked prior can lead to better uncertainty estimates.

Table 3: Results for multi-class image classification benchmarks. We report both loss and accuracy metrics for the training set and test set, together with the generalization loss and accuracy.

|  | Model | Train loss | Train acc. | Test loss | Test acc. | Gen. loss | Gen. acc. |
|---|---|---|---|---|---|---|---|
| MNIST | Softmax | 0.003 | 1.0 | 0.066 | 0.983 | 0.06 | 0.016 |
|  | MC-Dropout | 0.003 | 1.0 | 0.066 | 0.983 | 0.06 | 0.016 |
|  | BNN | 0.034 | 0.995 | 0.064 | 0.982 | **0.03** | **0.012** |
| Fashion | Softmax | 0.143 | 0.947 | 0.288 | 0.906 | 0.14 | 0.041 |
|  | MC-Dropout | 0.145 | 0.946 | 0.288 | 0.905 | 0.14 | 0.041 |
|  | BNN | 0.251 | 0.905 | 0.321 | 0.881 | **0.07** | **0.024** |
| CIFAR-10 | Softmax | 0.492 | 0.831 | 0.541 | 0.817 | 0.05 | 0.013 |
|  | MC-Dropout | 0.493 | 0.829 | 0.544 | 0.818 | 0.05 | 0.011 |
|  | BNN | 0.453 | 0.844 | 0.479 | 0.836 | **0.02** | **0.007** |

Table 4: Results for the out-of-distribution experiments. We report both loss and accuracy measures for the training set and test set, together with entropy values. We explore training on MNIST (MN), Fashion-MNIST (FMN), and CIFAR-10. Testing was done on MNIST (MN), Fashion-MNIST (FMN), NotMNIST, and SVHN. Reported losses and accuracies are computed using the out-of-distribution dataset.

| Model | In-dis. | Out-dis. | Train loss | Train acc. | Test loss | Test acc. | Train ent. | Test ent. |
|---|---|---|---|---|---|---|---|---|
| Softmax | FMN | MN | 8.33 | 0.113 | 8.21 | 0.114 | 0.42 | 0.41 |
| MC-Dropout | FMN | MN | 7.84 | 0.112 | 7.74 | 0.112 | 0.45 | 0.45 |
| BNN | FMN | MN | 4.64 | 0.105 | 4.57 | 0.108 | 0.78 | 0.74 |
| Softmax | FMN | NotMNIST | 8.64 | 0.155 | 7.78 | 0.152 | 0.43 | 0.41 |
| MC-Dropout | FMN | NotMNIST | 8.39 | 0.123 | 7.82 | 0.087 | 0.54 | 0.55 |
| BNN | FMN | NotMNIST | 4.70 | 0.085 | 4.59 | 0.1 | **0.89** | **0.92** |
| Softmax | CIFAR-10 | SVHN | 3.55 | 0.112 | 3.60 | 0.106 | 1.51 | 1.52 |
| MC-Dropout | CIFAR-10 | SVHN | 3.55 | 0.112 | 3.60 | 0.105 | 1.51 | 1.52 |
| BNN | CIFAR-10 | SVHN | 2.47 | 0.112 | 2.49 | 0.099 | **2.15** | **2.16** |

**In-Distribution Examples.** In the context of in-distribution examples we follow the suggestion of Guo et al. (2017) and calculate the Expected Calibration Error (ECE) and Maximum Calibration Error (MCE) for all three models. Figure 2b provides visual representation of the results. Results suggest that BNNs produce better calibrated outputs for all settings, with two exception of ECE for MNIST and MCE for CIFAR10.

**Out-of-Distribution Examples.** Next, we evaluated the uncertainty estimates using OOD examples. We apply a model trained using dataset A to OOD examples from dataset B. We trained models on MNIST, Fashion-MNIST and CIFAR-10 and assess prediction confidence using OOD examples from MNIST, Fashion-MNIST, NotMNIST (Cohen et al., 2017), and SVHN (Netzer et al., 2011). Results are summarized in Table 8. More OOD experiments using different dropout and prior rates can be found on Sections 8.3, 8.6 in the Appendix.

All models performed at the chance level ($\sim 10\%$ for 10 classes) for both OOD train and test sets. When considering the loss, we observe significantly higher values for the softmax and MC-Dropout models. These two findings imply that the softmax and MC-Dropout models are overly confident and tend to output a high probability for the max label. Hence, we measure the average entropy for all models. We expect BNNs to have higher entropy, due to the fact that it produces better uncertainty estimates, i.e., its' predictions for OOD samples are closer to a uniform distribution. Indeed, results reported in Table 8 confirm this intuition.

## 7  DISCUSSION AND FUTURE WORK

In the following study we present a new PAC-Bayesian generalization bound for learning a deep net using the NLL loss function. The proof relies on bounding the log-partition function using the squared norm of the gradients with respect to the input. Experimental validation shows that the resulting bound provides insight for better model optimization and prior distribution search. We demonstrate that learning the mean and STD for all parameters together with optimize prior over the parameters leads to better uncertainty estimates over the baselines and makes it harder to overfit.

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

## 8 APPENDIX

### 8.1 PAC-BAYESIAN BOUNDS FOR THE NEGATIVE LOG-LIKELIHOOD LOSS - PROOF

*Proof.* We begin by using the statistical independence of the training samples to decompose the following function:

$$
\mathbb{E}_{S \sim D^m}\left[e^{\lambda(L_D(w)-L_S(w))}\right] = e^{\lambda L_D(w)}\mathbb{E}_{S \sim D^m}\left[e^{\lambda \frac{1}{m}\sum_{i=1}^{m}(-\ell(w,x_i,y_i))}\right] \tag{10}
$$

$$
= e^{\lambda L_D(w)}\prod_{i=1}^{m}\mathbb{E}_{(x_i,y_i)\sim D}\left[e^{\frac{\lambda}{m}(-\ell(w,x_i,y_i))}\right] \tag{11}
$$

$$
= e^{\lambda L_D(w)}\left(\mathbb{E}_{(x,y)\sim D}\left[e^{\frac{\lambda}{m}(-\ell(w,x_i,y_i))}\right]\right)^m. \tag{12}
$$

Next we represent the moment generating function $M(\frac{\lambda}{m}) \triangleq \mathbb{E}_{(\hat{x},\hat{y})\sim D}\left[e^{\frac{\lambda}{m}(-\ell(w,\hat{x},\hat{y}))}\right]$ using $K(\alpha) \triangleq \frac{1}{\alpha}\log M(\alpha)$. The fundamental theorem of calculus asserts $K(\frac{\lambda}{m}) - K(0) = \int_0^{\lambda/m} K'(\alpha)d\alpha = \lambda/m \int_0^1 K'(\alpha)d\alpha$, where the last equality follows from a change of integration variable and the integral limits. $K'(\alpha)$ refers to the derivative at $\alpha$. We then compute $K'(\alpha)$ and $K(0)$:

$$
K'(\alpha) = \frac{\alpha M'(\alpha) - M(\alpha)\log M(\alpha)}{\alpha^2 M(\alpha)}, \tag{13}
$$

$$
K(0) = \lim_{\alpha \to 0^+}\frac{\log M(\alpha)}{\alpha} \overset{\text{l'Hopital}}{=} \frac{M'(0)/M(0)}{1} = M'(0) = -L_D(w). \tag{14}
$$

Concluding the Herbst argument we obtain the following equality:

$$
M\left(\frac{\lambda}{m}\right) = e^{\frac{\lambda}{m}K(\frac{\lambda}{m})} = e^{\frac{\lambda}{m}K(0)+(\frac{\lambda}{m})^2\int_0^1 K'(\alpha)d\alpha} = e^{-\frac{\lambda}{m}L_D(w)+(\frac{\lambda}{m})^2\int_0^1 \frac{\alpha M'(\alpha)-M(\alpha)\log M(\alpha)}{\alpha^2 M(\alpha)}d\alpha}. \tag{15}
$$

Combining Eq. (12) with Eq. (15) we derive:

$$
\mathbb{E}_{S \sim D^m}\left[e^{\lambda(L_D(w)-L_S(w))}\right] = e^{\lambda L_D(w)}\left(e^{-\frac{\lambda}{m}L_D(w)+(\frac{\lambda}{m})^2\int_0^1 \frac{\alpha M'(\alpha)-M(\alpha)\log M(\alpha)}{\alpha^2 M(\alpha)}d\alpha}\right)^m \tag{16}
$$

$$
= e^{\frac{\lambda^2}{m}\int_0^1 \frac{\alpha M'(\alpha)-M(\alpha)\log M(\alpha)}{\alpha^2 M(\alpha)}d\alpha}. \tag{17}
$$

Finally we apply the log-Sobolev inequality for Gaussian distributions (cf. Ledoux (2001), Chapter 2), as described in Eq. (6). To complete the proof we combine Eq. (6) with Eq. (17) to obtain:

$$
\mathbb{E}_{S \sim D^m}\left[e^{\lambda(L_D(w)-L_S(w))}\right] \leq e^{\frac{2\lambda^2}{m}\int_0^1 \frac{\mathbb{E}_{(x,y)\sim D}[e^{-\alpha\ell(w,x,y)}\alpha^2\|\nabla\ell(w,x,y)\|^2]}{\alpha^2 M(\alpha)}d\alpha} \tag{18}
$$

$$
= e^{\frac{2\lambda^2}{m}\mathbb{E}_{(x,y)\sim D}[\|\nabla\ell(w,x,y)\|^2\int_0^1 \frac{e^{-\alpha\ell(w,x,y)}}{M(\alpha)}d\alpha]} \tag{19}
$$

$\square$

### 8.2 APPLICATION TO LOGISTIC REGRESSION - PROOF

*Proof.* To apply Theorem 2 we start by realizing the gradient of $\log p(y|x,w)$ with respect to $x$. We denote by $w_y$ the $y-$th row of the parameter matrix $W$. Thus $\nabla_x \log p(y|w,x) = \sum_{\hat{y}} p(\hat{y}|x,w)(w_y - w_{\hat{y}})$. Using the convexity of the norm function we upper bound the gradient norm:

$$
\|\nabla_x \log p(y|w,x)\|^2 \leq \sum_{\hat{y}} p(\hat{y}|x,w)\|w_y - w_{\hat{y}}\|^2 \leq \|w_y\|^2 + \sum_{\hat{y}}\|w_{\hat{y}}\|^2 \leq 2\sum_{\hat{y}}\|w_{\hat{y}}\|^2 \tag{20}
$$

Next we use the fact that the gradient norm upper bound is independent of $x$ to simplify the moment generating function bound in Theorem 2. Since $\ell(w,x,y) = -\log p(\hat{y}|x,w)$, we use the bound in Eq. (20):

$$
\mathbb{E}_{(x,y)\sim D}\left[\|\nabla_x\ell(w,x,y)\|^2\int_0^1 \frac{e^{-\alpha\ell(w,x,y)}}{\mathbb{E}_{(\hat{x},\hat{y})\sim D}e^{\alpha(-\ell(w,\hat{x},\hat{y}))}}d\alpha\right] \leq 2\sum_{\hat{y}}\|w_{\hat{y}}\|^2\int_0^1 \frac{\mathbb{E}_{(x,y)\sim D}e^{-\alpha\ell(w,x,y)}}{\mathbb{E}_{(\hat{x},\hat{y})\sim D}e^{\alpha(-\ell(w,\hat{x},\hat{y}))}}d\alpha \tag{21}
$$

Thus we are able to simplify Theorem 2 as follows

$$\mathbb{E}_{w \sim q}[L_D(w)] \leq \mathbb{E}_{w \sim q}[L_S(w)] + \frac{\log \mathbb{E}_{w \sim p} e^{\frac{4\lambda^2}{m} \sum_{\hat{y}} \|w_{\hat{y}}\|^2} + KL(q||p) + \log(1/\delta)}{\lambda}. \tag{22}$$

Finally, we recall that $p$ is the prior density function $N(0, \sigma_p^2)$. Since the parameters are statistically independent, this expectation decomposes to its $kd$ parameters:

$$\mathbb{E}_{w \sim p} e^{\frac{4\lambda^2}{m} \sum_{\hat{y}} \|w_{\hat{y}}\|^2} = \prod_{\hat{y}=1}^{k} \prod_{i=1}^{d} \mathbb{E}_{w_{i,\hat{y}} \sim N(0, \sigma_p^2)} e^{\frac{4\lambda^2}{m} w_{i,\hat{y}}^2} \tag{23}$$

And the result follows from the fact $\mathbb{E}_{v \sim N(0, \sigma_p^2)} e^{\frac{4\lambda^2}{m} v^2} = \sqrt{\frac{m}{m - 8\lambda^2 \sigma_p^2}}$:

$$\int_{-\infty}^{\infty} \frac{1}{\sqrt{2\pi}\sigma_p} e^{-\frac{v^2}{2\sigma_p^2}} e^{\frac{4\lambda^2}{m} v^2} dv = \int_{-\infty}^{\infty} \frac{1}{\sqrt{2\pi}\sigma_p} e^{-v^2 \left(\frac{1}{2\sigma_p^2} - \frac{4\lambda^2}{m}\right)} dv = \frac{\rho}{\sigma_p} \int_{-\infty}^{\infty} \frac{1}{\sqrt{2\pi}\rho} e^{-\frac{v^2}{2\rho^2}} \tag{24}$$

for $\rho^2 = \frac{\sigma_p^2 m}{m - 8\lambda^2 \sigma_p^2}$. Whenever $\lambda \sigma_p \leq \sqrt{m/8}$ then $\sqrt{\frac{m}{m - 8\lambda^2 \sigma_p^2}} \leq \sqrt{2}$ and the result follows. $\qquad \square$

## 8.3 ARCHITECTURES

The architechtures described in this sub-section are used for the multi-class, binary, and uncertainty estimates experiments. We use multilayer perceptrons for the MNIST dataset, while we use convolutional neural networks (CNNs) for both Fashion-MNIST and CIFAR-10. A detailed description of the architectures is available in Table 5. We optimize the NLL loss function using SGD with a learning rate of 0.01 and a momentum value of 0.9 in all settings. We use mini-batches of size 128 and did not use any learning rate scheduling. For the MC-Dropout models we experienced with different weight decay values, however found that 1e-5 provides the best validation loss, hence choose this value.

Table 5: Model architectures. A description of the model architectures for MNIST, Fashion-MNIST and CIFAR-10. The numbers inside the parenthesis indicate the layer output dimension. For all models we use a ReLU activation function after each layer containing trainable weights.

| Dataset | No. of layers | Architecture |
|---|---|---|
| MNIST | 2 | FC(300) $\to$ FC(10) |
| Fashion-MNIST | 5 | Conv(10) $\to$ MaxPool(2) $\to$ Conv(20) $\to$ MaxPool(2) $\to$ FC(120) $\to$ FC(82) $\to$ FC(10) |
| CIFAR-10 | 6 | Conv(64) $\to$ Conv(64) $\to$ MaxPool(2) $\to$ Conv(64) $\to$ MaxPool(2) $\to$ Conv(64) $\to$ MaxPool(2) $\to$ FC(128) $\to$ FC(10) |

## 8.4 BINARY CLASSIFICATION

Experiments in this sub-section were conducted to show consistency with Dziugaite & Roy (2017). We follow the same setting in which we use the MNIST dataset, where we group digits [0, 1, 2, 3, 4] into label zero, and labels [5, 6, 7, 8, 9] into label one. All experiments in this subsection were conducted using multilayer perceptrons with one hidden layer consisting of 300 hidden neurons. We use the Rectified Linear Unit (ReLU) as our activation function (Glorot et al., 2011). We optimize the negative log-likelihood loss function using stochastic gradient descent (SGD) with a learning rate of 0.1 and a momentum value of 0.9. We did not use any learning rate scheduling. SGD is run in mini-batches of size 128. Each model was trained for 20 epochs.

We compared BNN to softmax models with dropout (Srivastava et al., 2014) rates chosen from the set $\{0.0, 0.3, 0.5\}$. Hereby, a dropout with a rate of 0.0 means no dropout at all. In addition to the training set and test set loss and accuracy, we measure the *generalization loss*, while setting $L_D(w)$ to be the average test set loss. In the same manner, we measure the *generalization accuracy*, while using the zero-one loss instead of the negative log-likelihood loss. Table 6 summarizes the results. All models achieve comparable accuracy levels, however the softmax models suffer from larger generalization errors both in terms of loss and accuracy. Notice, as expected, using higher Bernoulli-dropout rates mitigates the generalization gap.

Table 6: Results for experiments on a binary classification variant of MNIST. We report both loss and accuracy metrics for the training set and test set, together with the generalization loss and accuracy. Additionally, we report the KL-divergence value for the BNN models. We compare BNN to softmax models trained with Bernoulli dropout. Dropout rates and $\sigma_p$ values are in parenthesis next to model name, e.g., 'softmax (0.3)' refers to a softmax model with Bernoulli dropout of rate 0.3.

| Model | Train loss | Train acc. | Test loss | Test acc. | KL | Gen. loss | Gen. acc. |
|---|---|---|---|---|---|---|---|
| Softmax (0.0) | 0.001 | 1.0 | 0.076 | 0.983 | - | 0.0746 | 0.017 |
| Softmax (0.3) | 0.011 | 1.0 | 0.055 | 0.983 | - | 0.0439 | 0.013 |
| Softmax (0.5) | 0.021 | 0.993 | 0.059 | 0.982 | - | 0.0379 | 0.010 |
| BNN (0.1) | 0.046 | 0.984 | 0.064 | 0.979 | 4950 | **0.0187** | **0.005** |
| BNN (0.3) | 0.047 | 0.983 | 0.081 | 0.975 | 4196 | 0.0331 | 0.008 |

## 8.5 MULTI-CLASS CLASSIFICATION

Here we report results for multi-class classification for BNN and the baselines. Table 7 summarizes the results. The main purpose of these additional experiments is to explore more dropout and $\sigma_p$ values for different models.

Table 7: Results for multi-class image classification benchmarks. We report both loss and accuracy metrics for the training set and test set, together with the generalization loss and accuracy. Additionally, we report the KL-divergence for the BNN models. We compare BNN to softmax models and MC-Dropout models, trained with Bernoulli dropout. Dropout rates and $\sigma_p$ values are in parenthesis next to the model name, e.g., 'softmax (0.5)' refers to a softmax model with Bernoulli dropout of rate 0.5.

| Model | Train loss | Train acc. | Test loss | Test acc. | KL | Gen. loss | Gen. acc. |
|---|---|---|---|---|---|---|---|
| | | | **MNIST** | | | | |
| Softmax (0.0) | 0.0003 | 1.0 | 0.0699 | 0.983 | - | 0.070 | 0.02 |
| Softmax (0.1) | 0.0008 | 1.0 | 0.0651 | 0.984 | - | 0.064 | 0.016 |
| Softmax (0.5) | 0.0103 | 1.0 | 0.0682 | 0.983 | - | 0.058 | 0.014 |
| Softmax(0.7) | 0.012 | 1.0 | 0.0658 | 0.983 | - | 0.054 | 0.013 |
| Softmax(0.9) | 0.1026 | 0.97 | 0.1385 | 0.962 | - | 0.036 | 0.008 |
| MC-Dropout (0.1) | 0.0009 | 1.0 | 0.0651 | 0.984 | - | 0.064 | 0.016 |
| MC-Dropout (0.5) | 0.0111 | 1.0 | 0.0678 | 0.982 | - | 0.057 | 0.015 |
| MC Dropout(0.7) | 0.0178 | 0.99 | 0.0781 | 0.979 | - | 0.0603 | 0.016 |
| MC Dropout(0.9) | 0.1373 | 0.96 | 0.1771 | 0.953 | - | 0.0398 | 0.007 |
| BNN (0.3) | 0.0448 | 0.986 | 0.0647 | 0.98 | 4251 | 0.02 | 0.006 |
| | | | **Fashion-MNIST** | | | | |
| Softmax (0.0) | 0.0159 | 0.995 | 0.6415 | 0.896 | - | 0.62 | 0.098 |
| Softmax (0.1) | 0.0899 | 0.966 | 0.3644 | 0.902 | - | 0.27 | 0.064 |
| Softmax (0.5) | 0.1941 | 0.927 | 0.3106 | 0.895 | - | 0.12 | 0.032 |
| Softmax(0.7) | 0.2361 | 0.915 | 0.333 | 0.892 | - | 0.097 | 0.023 |
| Softmax(0.9) | 0.8366 | 0.643 | 0.866 | 0.639 | - | 0.03 | 0.004 |
| MC-Dropout (0.1) | 0.0913 | 0.965 | 0.3587 | 0.901 | - | 0.27 | 0.064 |
| MC-Dropout (0.5) | 0.1973 | 0.926 | 0.3097 | 0.897 | - | 0.11 | 0.029 |
| MC Dropout(0.7) | 0.2535 | 0.913 | 0.3387 | 0.887 | - | 0.0852 | 0.026 |
| MC Dropout(0.9) | 0.908 | 0.608 | 0.9318 | 0.602 | - | 0.0238 | 0.006 |
| BNN (0.3) | 0.1841 | 0.932 | 0.2640 | 0.904 | 4869 | 0.08 | 0.028 |
| | | | **CIFAR-10** | | | | |
| Softmax (0.0) | 0.3361 | 0.881 | 0.4974 | 0.841 | - | 0.16 | 0.039 |
| Softmax (0.1) | 0.4170 | 0.858 | 0.5064 | 0.828 | - | 0.09 | 0.351 |
| Softmax (0.5) | 0.5299 | 0.821 | 0.5712 | 0.804 | - | 0.04 | 0.017 |
| Softmax(0.7) | 0.4366 | 0.852 | 0.5494 | 0.815 | - | 0.113 | 0.037 |
| Softmax(0.9) | 1.2778 | 0.495 | 1.3491 | 0.499 | - | 0.071 | -0.004 |
| MC-Dropout (0.1) | 0.4189 | 0.856 | 0.5071 | 0.830 | - | 0.09 | 0.026 |
| MC-Dropout (0.5) | 0.5293 | 0.821 | 0.5711 | 0.807 | - | 0.04 | 0.014 |
| MC Dropout(0.7) | 0.4595 | 0.843 | 0.5878 | 0.811 | - | 0.128 | 0.033 |
| MC Dropout(0.9) | 1.3045 | 0.483 | 1.3935 | 0.482 | - | 0.089 | 0.001 |
| BNN (0.3) | 0.5643 | 0.807 | 0.5699 | 0.806 | 18237 | 0.005 | 0.001 |

## 8.6 UNCERTAINTY ESTIMATES

Additional experimental results for out-of-distribution examples can be found in Table 8.

Table 8: Results for the out-of-distribution experiments. We report both loss and accuracy measures for the training set and test set, together with entropy values. We explore training on MNIST (MN), Fashion-MNIST (FMN), and CIFAR-10. Testing was done on MNIST (MN), Fashion-MNIST (FMN), NotMNIST, and SVHN. Reported losses and accuracies are computed using the out-of-distribution dataset.

| Model | In-dis. | Out-dis. | Train loss | Train acc. | Test loss | Test acc. | Train ent. | Test ent. |
|---|---|---|---|---|---|---|---|---|
| Softmax (0.0) | FMN | MN | 10.32 | 0.136 | 10.04 | 0.146 | 0.34 | 0.34 |
| Softmax (0.1) | FMN | MN | 9.41 | 0.114 | 9.28 | 0.116 | 0.42 | 0.41 |
| Softmax (0.5) | FMN | MN | 8.16 | 0.114 | 7.97 | 0.114 | 0.42 | 0.41 |
| MC-Dropout (0.1) | FMN | MN | 9.12 | 0.113 | 8.99 | 0.113 | 0.44 | 0.43 |
| MC-Dropout (0.5) | FMN | MN | 7.68 | 0.113 | 7.50 | 0.113 | 0.46 | 0.45 |
| BNN (0.3) | FMN | MN | 5.31 | 0.096 | 5.22 | 0.098 | 0.79 | 0.75 |
| Softmax (0.0) | MN | FMN | 14.94 | 0.109 | 15.03 | 0.104 | 0.28 | 0.29 |
| Softmax (0.1) | MN | FMN | 16.27 | 0.098 | 16.33 | 0.092 | 0.29 | 0.29 |
| Softmax (0.3) | MN | FMN | 18.07 | 0.094 | 18.15 | 0.089 | 0.30 | 0.31 |
| Softmax (0.5) | MN | FMN | 18.78 | 0.077 | 18.89 | 0.075 | 0.38 | 0.38 |
| MC (0.1) | MN | FMN | 16.28 | 0.098 | 16.34 | 0.093 | 0.29 | 0.29 |
| MC (0.3) | MN | FMN | 18.09 | 0.093 | 18.16 | 0.090 | 0.31 | 0.31 |
| MC (0.5) | MN | FMN | 18.83 | 0.076 | 18.93 | 0.076 | 0.37 | 0.38 |
| BNN (0.1) | MN | FMN | 6.69 | 0.102 | 6.70 | 0.099 | 0.72 | 0.73 |
| BNN (0.3) | MN | FMN | 8.67 | 0.069 | 8.71 | 0.069 | 0.55 | 0.57 |
| Softmax (0.0) | MN | NotMNIST | 13.48 | 0.078 | 13.84 | 0.085 | 0.27 | 0.27 |
| Softmax (0.1) | MN | NotMNIST | 15.45 | 0.061 | 16.03 | 0.065 | 0.23 | 0.24 |
| Softmax (0.3) | MN | NotMNIST | 19.38 | 0.071 | 19.25 | 0.085 | 0.20 | 0.18 |
| Softmax (0.5) | MN | NotMNIST | 21.61 | 0.063 | 21.89 | 0.048 | 0.18 | 0.18 |
| MC (0.1) | MN | NotMNIST | 15.46 | 0.061 | 16.05 | 0.065 | 0.23 | 0.24 |
| MC (0.3) | MN | NotMNIST | 19.44 | 0.072 | 19.23 | 0.076 | 0.20 | 0.18 |
| MC (0.5) | MN | NotMNIST | 21.73 | 0.062 | 21.92 | 0.053 | 0.19 | 0.19 |
| BNN (0.1) | MN | NotMNIST | 5.67 | 0.087 | 5.52 | 0.085 | 0.79 | 0.74 |
| BNN (0.3) | MN | NotMNIST | 7.71 | 0.060 | 7.99 | 0.070 | 0.57 | 0.57 |
| Softmax (0.0) | FMN | NotMNIST | 12.46 | 0.152 | 11.92 | 0.13 | 0.29 | 0.27 |
| Softmax (0.1) | FMN | NotMNIST | 9.71 | 0.080 | 9.71 | 0.074 | 0.32 | 0.38 |
| Softmax (0.5) | FMN | NotMNIST | 8.79 | 0.122 | 8.22 | 0.087 | 0.5. | 0.52 |
| MC-Dropout (0.1) | FMN | NotMNIST | 9.45 | 0.079 | 9.45 | 0.076 | 0.34 | 0.4 |
| MC-Dropout (0.5) | FMN | NotMNIST | 8.23 | 0.156 | 7.47 | 0.157 | 0.46 | 0.44 |
| BNN (0.3) | FMN | NotMNIST | 4.74 | 0.123 | 4.72 | 0.11 | 0.79 | 0.81 |
| Softmax (0.0) | CIFAR-10 | SVHN | 4.93 | 0.115 | 5.13 | 0.109 | 0.94 | 0.96 |
| Softmax (0.1) | CIFAR-10 | SVHN | 3.78 | 0.123 | 3.86 | 0.115 | 1.51 | 1.54 |
| Softmax (0.5) | CIFAR-10 | SVHN | 3.11 | 0.121 | 3.19 | 0.131 | 1.73 | 1.75 |
| MC-Dropout (0.1) | CIFAR-10 | SVHN | 3.79 | 0.123 | 3.87 | 0.115 | 1.50 | 1.53 |
| MC-Dropout (0.5) | CIFAR-10 | SVHN | 3.10 | 0.122 | 3.17 | 0.132 | 1.74 | 1.77 |
| BNN (0.3) | CIFAR-10 | SVHN | 3.45 | 0.091 | 3.54 | 0.094 | 1.71 | 1.75 |

## 8.7 FIGURES

Additional figures for various experiments.

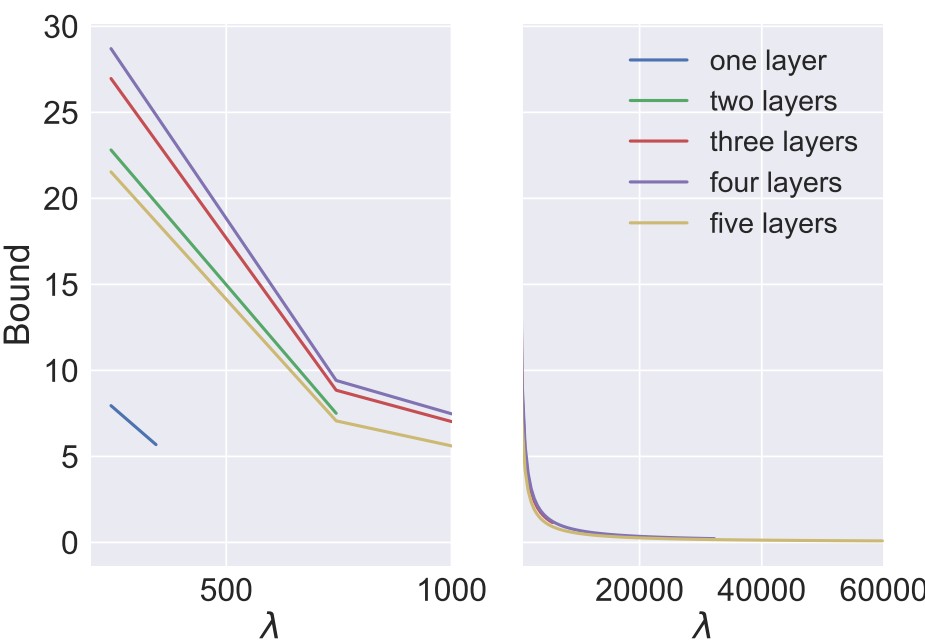

Figure 3: Analysis of the proposed bound as a function network depth. We report the the generalization bound as a function of $\lambda$ for different deep net depth levels using the Fashion-MNIST dataset.

