# OpenReview forum: "PAC-Bayesian Neural Network Bounds"
_ICLR.cc/2020/Conference — Reject_

### Official Review · AnonReviewer3 · 2019-10-24
**Official Blind Review #3**

**Rating:** 6

**Review:**

The paper offers PAC-generalization bounds for Bayesian Neural Networks based on a previous result by Alquier et al. (Theorem 1) which connects the generalization gap to the log partition function of the same gap for the prior distribution on the learned parameters (which is identical to the ELBO bound used in Bayesian neural networks for NLL loss). Due to the fact that the optimal bound occurs for the true posterior, the PAC-bayesian bounds offer a novel interpretation as an objective for BNNs.

The authors note that the log partition function can in general be easily unbounded for loss functions based on NLL (as in the BNN case); their result shows that if the norm of the gradient is bounded, that is enough to bound the overall generalization gap.

While this appears to be a technically impressive feat, the the assumptions involved in Theorem 2 seem significant (probably unavoidable for a theoretically tractable statement). Primarily, the conditional of x given y is Gaussian/log-concave (or at least unimodal, more generally ) but the motivation is based on deep neural networks (for why the gradient is bounded).

The authors also specialize their bound to the case of logistic regression. Interestingly, the gap in this case has an additive term proportional to the product of the label cardinality and the input dimension (I'm not sure whether how significant this is in terms of tightness).

 In experiments, the authors explore and analyze the tightness of the proposed bounds for various hyperparameters like the variance of the weights prior.

They also perform an exhaustive comparison of the BNN models against non-bayesian alternatives, but it is not clear how the new contributions from the generalization bounds are relevant to the results in, say Section 6.2

**Experience Assessment:**

I do not know much about this area.

**Review Assessment: Checking Correctness Of Derivations And Theory:**

I assessed the sensibility of the derivations and theory.

**Review Assessment: Checking Correctness Of Experiments:**

I assessed the sensibility of the experiments.

**Review Assessment: Thoroughness In Paper Reading:**

I made a quick assessment of this paper.

---

> ### Author Response · Authors · 2019-11-10
> **Authors response**
>
> We thank the reviewer for taking the time to review our paper.
> Regarding the Gaussian/log-concave assumption:
> We believe such an assumption is reasonable for various input domains (especially compared to the assumptions in previous work, such as linearity, convexity, bounded loss, etc.).
>
> Regarding the connection between the bound and the results:
> The premise is that using the proposed bound we can choose better prior distribution over the weights, which results in better posterior distribution over the weights which translates to better generalization and better uncertainty estimates.

---

### Official Review · AnonReviewer4 · 2019-10-29
**Official Blind Review #4**

**Rating:** 3

**Review:**

Summary:
This paper proposes a PAC-Bayesian generalization bound for Bayesian neural networks. The author discuss that earlier generalization bounds hold for bounded loss function, whereas the proposed generalization bound considers negative log likelihood loss which can be unbounded. Therefore, earlier results do not hold here.

The technical approach used is along lines of PAC Bayes framework and specifically for this loss function which requires bounding the log-partition function, the authors follow the Herbst Argument for bounding the log-partition function.

Contribution:
The paper uses straight forward PAC Bayes approach and the only bottleneck is bounding the log-partition function for which the authors use an earlier result (Herbst Argument for bounding the log-partition function. )

Significance:
My biggest concern with this work is its significance. As we know classification loss is bounded and for regression loss, as long as we have bounded input and a Lipschitz function corresponding to the NN, the output is bounded. Also as authors mention, there have been two earlier results covering other unbounded loss functions. Therefore, I do not feel that extending those results to NLL is a good enough contribution. Especially since the extension uses a known approach  (Herbst Argument for bounding the log-partition function. )

Experiments:
* From the explanations, it seems each architecture is trained once, which is not acceptable. How can one refute the effect of a specific initial value? A good scientific practice entails having mean and variance bar for different values or at least repeating the experiment multiple times and reporting the avg.
* According to the paper, the architectures in table 2, fig 1 are made by keeping the number of parameters roughly the same. Then the authors increase the depth. Note that to keep the # of parameters the same, they have to decrease the width as they increase the depth. Therefore, this cannot be a correct analysis of effect of the depth. As depth is not the only parameter that is changed between the architectures.


Writing:
The writing is overall ok which some vague places such as
*first page, last paragraph, line 1: ".. our PAC Bayesian bound is tightest when we learn..." the authors do not discuss what is the space of options for the bound and only mention the case when it is tightest. Therefore the claim is confusing
*first page, last paragraph, last line: "..better uncertainty than the baseline". The authors do not specify the baseline which makes this whole claim vague.

Title:
This is a minor issue but worth mentioning. The title is vague and confusing. In the first read one might think this paper provides PAC-Bayesian bounds for usual NNs (which has been considered and written about many times in the literature). The authors should mention that the considered networks are Bayesian NNs.

**Experience Assessment:**

I have published one or two papers in this area.

**Review Assessment: Checking Correctness Of Derivations And Theory:**

I assessed the sensibility of the derivations and theory.

**Review Assessment: Checking Correctness Of Experiments:**

I assessed the sensibility of the experiments.

**Review Assessment: Thoroughness In Paper Reading:**

I read the paper thoroughly.

---

> ### Author Response · Authors · 2019-11-10
> **Authors response**
>
> Regarding the significance concern:
> We kindly disagree with the reviewer that the classification loss is always bounded. While the zero-one loss function is bounded, it cannot be used for training. The hinge-loss can be bounded under the assumption of bounded inputs and Lipschitz function corresponding to the model.
> While computing the Lipschitz constant in NP-Hard for deep nets (https://arxiv.org/pdf/1805.10965.pdf), The Lipschitz constant increases in the depth of the network, while our bound holds for unbounded input and non-Lipschitz functions, and decreases in the depth of the network. Hence, Lipschitz-type bounds are very crude, e.g., they do not distinguish between different activation functions with the same Lipschitz constant. For example, our bound achieves fast-rate (1/m) for deep networks, and as far as we know this is the only bound that achieves it under this setting.
>
> Regarding previous work for unbounded loss functions:
> We kindly ask the reviewer to clarify. As far as we know, the other results for unbounded loss functions consider linear models, while our bound considers deep networks, which is certainly a significant difference. Can you please elaborate on the difference, which we might overlooked. We also point out that although our treatment holds for any (almost anywhere) smooth loss function, the NLL loss is the most widely used loss these days, and this extension only is probably of interest to the community.
>
> Regarding the Herbst argument:
> To the best of our knowledge, the Herbst argument was never used in PAC-Bayesian setting (nor in any generalization bound). Moreover, our use of the Herbst argument is different than previous works in functional analysis (e.g., Gross 1975, Ledoux 2001) that use it for Lipschitz functions, while our work use it for *non-Lipschtiz* functions.
>
> Regarding the experiments:
> We agree with the reviewer that depth is not the only parameter that is being changed, however, since the goal of this experiment was to explore the effect of the new MGF bound (the complexity of the model), which is dominated by the norm of the gradients w.r.t the input to the model, keeping the number of model parameters roughly the same is essential in order to create comparable KL values between the models. Increasing the number of parameters may cause higher KL values which will greatly affect the generalization bound.
> Regarding the repeated experiments, we will add mean and std measures for all experiments in the final manuescript.
>
> Regarding writing and title:
> We will clarify all above comments for the final manuscript

---

### Official Review · AnonReviewer2 · 2019-10-31
**Official Blind Review #2**

**Rating:** 3

**Review:**

This paper suggests a PAC-Bayesian bound for negative log-likelihood loss function. Many PAC-Bayesian bounds are provided for bounded loss functions but as authors point out, Alquier et al. (2016) and Germain et al. (2016) extend them to  unbounded loss functions. I have two major concerns regarding this paper:

1- Technical contribution: Since Alquier et al. (2016) has already introduced PAC-Bayesian bounds for the hinge-loss, I think the technical contributions of this paper is not significant enough for the publication. Moreover, the particular format of the bound in Theorem 2 is problematic since the right hand side depends on the data-distribution. When presenting the generalization bound, we really want the right hand side to be independent of the distribution (given the training set) and that is the whole point of calculating the generalization bounds. In particular, I don't see why inequality (1) is any better than inequality (2).

2- Experiments: The main issue with the correlation analysis done in Section 6 is that authors only change depth of the networks and then check the correlation of the generalization bound to the test error. The problem is that in all those networks deeper ones generalize better so it is not clear that the correlation is due to a direct relationship to generalization or a direct relationship to depth. For example, if we take 1/depth as a measure, it would correlate very well with generalization in all these experiments but 1/depth is definitely not the right complexity measure or generalization bound. To improve the evaluation, I suggest varying more hyperparameters to avoid the above issue.


***************************

After rebuttals:

Unfortunately, my concerns are not addressed adequately by authors. Therefore, my evaluation remains the same.

**Experience Assessment:**

I have published in this field for several years.

**Review Assessment: Checking Correctness Of Derivations And Theory:**

I assessed the sensibility of the derivations and theory.

**Review Assessment: Checking Correctness Of Experiments:**

I assessed the sensibility of the experiments.

**Review Assessment: Thoroughness In Paper Reading:**

I read the paper at least twice and used my best judgement in assessing the paper.

---

> ### Author Response · Authors · 2019-11-10
> **Authors response**
>
> Regarding comparison with Alquier et al. (2016):
> We kindly disagree with this assessment, which treats deep networks as linear models. Alquier et al. proved their bound for the hinge loss for the linear case which does not hold for deep networks. Our bound holds for any (almost everywhere) smooth loss function, which includes both linear models and deep neural models with the hinge loss. To better emphasize the above point, we presented the applicability of our bound to the linear case of the NLL loss function, which is the logistic regression case. We were focused on the NLL loss function for deep networks due to its extensively usage nowadays. Moreover, our bound achieves fast-rates (1/m), as presented in Figure 1(b), for deep networks while maintaining fixed prior variance, in contrast to Alquier et al.
>
> Regarding theorem 2 format:
> Traditionally, this is the view when presenting a generalization bound, a view which we also shared for a long time. Over time we came to the conclusion that this limits our understanding of Bayeisan deep networks. Our bound utilizes the distribution D and the prior distribution over the weights p, both distributions do not rely on the training data S.
>
> Regarding why inequality (1) is better than inequality (2):
> We appreciate the reviewer allowed a specific example to emphasize our contribution. Inequality (1) cannot be computed since it requires to sample training set S multiple times and evaluate the Moment Generating Function (MGF) with respect to it. Since S dimension is about 60k (MNIST/Fashion-MNIST/CIFAR), one would need an infinite amount of time to estimate the MGF.
> Instead, one can use the independence assumption of S to decompose the MGF (similar to Equation (12) in the appendix). We tried to go this path, and since it estimates the expectation of an exponential function, it reaches infinity (nan) for lambda that is approximately ~15-20 on MNIST (recall, m=60k), i.e., the rate of this approach is much lower than the conventional $\sqrt{m}$. In contrast, since our bound relies on gradients in the log-domain, we can show fast-rate bound (i.e., lambda = 60k) when the network turns deeper (see Figure 1b).
>
> Regarding the depth experiments:
> It is not true that deeper networks generalize better in all cases. The generalization bound consists of two terms, the complexity of the model (our new MGF bound, which is dominated by the norm of the gradients) and the complexity of the learning (which is controlled by the KL-divergence). The goal of this experiment was to explore the effect of the new MGF bound (the complexity of the model). Hence, we kept the number of model parameters roughly the same in order to create comparable KL values between the models. Increasing the number of parameters may cause higher KL values which will greatly affect the generalization bound.
> We agree that if we keep the KL-divergence fixed, then the MGF decreases with the depth, but this varies with the components that are being used (convolutional layers, skip-connections, fully connected, etc.). According to Figure 1(a), we see that it is not the depth that matters but the norm of the gradient w.r.t the input, which is exactly what’s dominates the MGF bound.

---

> > ### Comment · AnonReviewer2 · 2019-11-13
> > **Thanks for the response**
> >
> > Thank you for your response.
> >
> > I'm not convinced by your argument about the distribution dependence. I believe it is OK to use the training set information on the right hand side but since we don't have access to the distribution, it does not make sense to have distribution dependence on the right hand side.
> >
> > About the depth experiments: My observation is that your theoretical bound is highly correlated with depth and it is not clear if the correlation with generalization is because of that or not. In particular, in Table 2, your bound is always lower for deeper networks; however, it doesn't always give the right order for generalization. For example, four layer neural nets perform worse than three layer ones on MNIST and Boston but your bound is lower for four layer networks.

---

> > > ### Author Response · Authors · 2019-11-15
> > > **Response**
> > >
> > > We would like to thank you for clarifying your concern.
> > > The distribution dependence concern can be mitigated by standard concentration of measure bounds (e.g., Chebyshev's inequality), while assuming that the variance of the gradient norm is bounded (which holds since the gradients decay very fast).
> > > Regarding the experiments. The MGF is correlated with the gradients norm (fig. 1a) which may be correlated with the depth, depends on model architecture and the chosen transfer functions. However, we would like to clarify that the bound (computed in Table 2) is both the expected gradient norm and the KL term, hence, there is an interesting balance between both of them which causing lower generalization bounds for deeper models.

---

### Decision · Program_Chairs · 2019-12-19

**Decision:**

Reject

**Comment:**

This paper proposes PAC_Bayesian bounds for negative log-likelihood loss function. A few reviewers raised concerns around 1) distinguish their contributions better from prior work (eg Alquier). 2) confounders in their experiments. Both reviewers agreed that the paper, as it is written, does not provide sufficient evidence of significance. In addition, experiments shown in the paper varies two things - # parameters (therefore expressiveness and potential generalizability) and depth at each setting. As pointed out, this isn’t right - in order to capture the effect, one has to control for all confounders carefully. Another concerned raised were around Theorem 2 - that it contains data-distribution on the right hand side, which isn’t all that useful to calculate generalization bounds (we don’t have access to the distribution). We highly encourage authors to take another cycle of edits to better distinguish their work from others before future submissions.